Broad spectrum pesticide application alters natural enemy communities and may facilitate secondary pest outbreaks

Hill Matthew P. matthill@protonmail.com 1
Macfadyen Sarina 1
Nash Michael A. 2
1 Agriculture & Food, CSIRO , Canberra , Australian Capital Territory , Australia
2 School of Agriculture, Food and Wine, University of Adelaide , Waite Campus , South Australia , Australia
Sandhu Harpinder
Electronic publication date: 2017 Dec 19
Publication date: 2017
Volume: 5
Electronic Location ID: e4179
Received 2017 Aug 6; Accepted 2017 Nov 29
Copyright: ©2017 Hill et al.
Copyright year: 2017
Copyright holder: Hill et al.
License: This is an open access article distributed under the terms of the Creative Commons Attribution License, which permits unrestricted use, distribution, reproduction and adaptation in any medium and for any purpose provided that it is properly attributed. For attribution, the original author(s), title, publication source (PeerJ) and either DOI or URL of the article must be cited.
License URL: https://creativecommons.org/licenses/by/4.0/

Keywords: Organophosphate, Community ecology, Pesticide, Secondary outbreak, Pestsuppression

Funding: Grains Research & Development Corporation CSE00059 This work was supported by the Grains Research & Development Corporation (CSE00059). The funders had no role in study design, data collection and analysis, decision to publish, or preparation of the manuscript.

==============================
Background

Pesticide application is the dominant control method for arthropod pests in broad-acre arable systems. In Australia, organophosphate pesticides are often applied either prophylactically, or reactively, including at higher concentrations, to control crop establishment pests such as false wireworms and earth mite species. Organophosphates are reported to be disruptive to beneficial species, such as natural enemies, but this has not been widely assessed in Australian systems. Neither has the risk that secondary outbreaks may occur if the natural enemy community composition or function is altered.

Methods

We examine the abundance of ground-dwelling invertebrate communities in an arable field over successive seasons under rotation; barley, two years of wheat, then canola. Two organophosphates (chlorpyrifos and methidathion) were initially applied at recommended rates. After no discernible impact on target pest species, the rate for chlorpyrifos was doubled to elicit a definitive response to a level used at establishment when seedling damage is observed. Invertebrates were sampled using pitfalls and refuge traps throughout the experiments. We applied measures of community diversity, principal response curves and multiple generalised linear modelling techniques to understand the changes in pest and natural enemy communities.

Results

There was large variability due to seasonality and crop type. Nevertheless, both pest (e.g., mites and aphids) and natural enemy (e.g., predatory beetles) invertebrate communities were significantly affected by application of organophosphates. When the rate of chlorpyrifos was increased there was a reduction in the number of beetles that predate on slug populations. Slugs displayed opposite trends to many of the other target pests, and actually increased in numbers under the higher rates of chlorpyrifos in comparison to the other treatments. Slug numbers in the final rotation of canola resulted in significant yield loss regardless of pesticide application.

Discussion

Organophosphates are a cost-effective tool to control emergent pests in broad-acre arable systems in Australia. We found risks associated with prophylactic application in fields under rotation between different crop types and significant changes to the community of pests and natural enemy. Disrupting key predators reduced effective suppression of other pests, such as slugs, and may lead to secondary outbreaks when rotating with susceptible crops such as canola. Such non-target impacts are rarely documented when studies focus on single-species, rather than community assessments. This study represents a single demonstration of how pesticide application can lead to secondary outbreaks and reinforces the need for studies that include a longer temporal component to understand this process further.

Introduction

Pesticides predominate management options for control of invertebrate pests in many parts of the world (Thomson & Hoffmann, 2006; Guedes et al., 2016). The most widely used pesticide class in Australia is organophosphates, with ∼5,000 tonnes applied annually across agricultural systems in 2002 (Radcliffe, 2002). Despite an increase in use of pesticides, crop losses due to pests have remained largely unchanged for 30–40 years (Altieri & Nicholls, 2004). Beyond the target pests, broad-spectrum pesticides (that kill insects and mites indiscriminately) may affect non-target invertebrate species (Readshaw, 1975), including causing reductions in natural enemy population abundance and activity (e.g., Wilson, Bauer & Lally, 1998; Wilson, Bauer & Lally, 1999), and competition between pest species (known as competitive release, Zeilinger, Olson & Andow, 2016). Assays of invertebrates against weathered residues have shown the persistence of pesticides might play an important part in their negative impacts on natural enemies in the field (Grundy et al., 2000).

A potential outcome of frequent broad-spectrum pesticide use is the emergence of pests not controlled by the pesticides but benefiting from reduced mortality from natural enemies and/or competitive release, commonly known as secondary pests (Dutcher, 2007; Gross & Rosenheim, 2011; Steinmann, Zhang & Grant, 2011). Reporting secondary pest outbreaks is challenging as they may also be caused by other mechanisms, which inherently makes it difficult to determine how frequently pesticide-use results in this outcome (Gross & Rosenheim, 2011). In cotton fields, it was estimated that 20% of late-season pesticide costs were attributable to secondary pest outbreaks caused by early-season pesticide applications for Lygus pests (Gross & Rosenheim, 2011). Higher numbers of cotton aphids, Aphis gossypii Glover and spider mites, Tetranychus urticae Koch were found in cotton fields that received early-season applications of insecticides against Helicoverpa spp. (Wilson, Bauer & Lally, 1998; Wilson, Bauer & Lally, 1999). Understanding interactions between resident invertebrate communities and pesticides will help us predict when secondary pest outbreaks are likely to occur, and lead to more informed pest control decision-making.

One standardised approach for assessing non-target impacts of pesticides is the International Organization for Biological and Integrated Control—Pesticides and Beneficial Organisms (IOBC) ratings system (Hassan et al., 1985). This approach has identified a range of toxic and harmful effects of broad-spectrum pesticides on a number of non-target invertebrate species, particularly natural enemies. As this ratings system focuses on standardised sets of “representative” organisms, it does not consider the specific context in which pesticides are being applied, the rate at which they are applied nor the cumulative effects of multiple chemical applications across a season (Nash, Thomson & Hoffmann, 2008a). This means that the diverse range of sub-lethal effects are not assessed (Stark, Banks & Acheampong, 2004). Subsequently, more bioassays under field conditions are needed to incorporate the dynamic interaction between pest populations and their natural enemy communities (Thomson & Hoffmann, 2007) and the environmental context at the time of application. While such studies are rare for examining repeated pesticide applications in arable systems, community-level analyses to examine the effects of genetically modified crops (e.g., Bt cotton) on non-target species are more commonplace (Naranjo, 2005; Whitehouse, Wilson & Fitt, 2005; Rose & Dively, 2007). This suggests that such methods should be transferable to examine repeated pesticide applications on communities of pests and natural enemies within Australian arable systems.

In Australian broad-acre grains the pest management practitioners are primarily concerned with pesticide efficacy, crop phytotoxicity and cost; seldom are broader impacts of pesticides included in decision-making (Van der Werf, 1996; Umina et al., 2015). As such, more expensive selective pesticides are not favoured. Two broad-spectrum organophosphate pesticides, methidathion and chlorpyrifos, are commonly used to control invertebrate pests. Methidathion is typically used to control earth mites and lucerne flea in emerging canola crops, and chlorpyrifos is used to control mite and wireworm larvae around sowing (Gu, Fitt & Baker, 2007). Chlorpyrifos is applied for the control of pests such as earwigs, isopods (Armadillidiidae) and millipedes (Portuguese millipede, Ommatoiulus moreleti Lucas, 1860) (MA Nash, pers. obs., 2014), despite not being registered specifically to control those pests. A reduced application rate of broad-spectrum pesticides may lessen the impact on natural enemies, but still remain efficacious against pests (e.g., Wiles & Jepson, 1995; Wilson, Bauer & Lally, 1998). However, when growers fail to achieve what they consider to be adequate pest control they often respond by applying higher rates of pesticides, especially for high yielding crops that are likely to still generate a significant profit despite the added input costs (Edwards et al., 2008). Repeated applications of broad-spectrum pesticides to control typical pest species is common in broad-acre crops, in particular canola (Gu, Fitt & Baker, 2007) and pulses (Murray, Clarke & Ronning, 2013). There are few economic thresholds for many pest species common in Australian grain crops (but see Arthur, Hoffmann & Umina, 2015), therefore growers cannot often relate the pest numbers observed in a field to likely yield losses and adjust pesticide application accordingly (e.g., aphids; Valenzuela & Hoffmann, 2015). The outcome is that pesticides are often applied prophylactically or in response to some observed crop damage that may or may not result in yield loss.

Since the late 1990s, a number of exotic slug species have also emerged as pests of canola at the crop establishment stage across the high rainfall (>500 mm growing season rainfall) zones of southern Australia (Nash, Thomson & Hoffmann, 2007). Two common species, Deroceras reticulatum Müller, 1774 and Milax gagates Draparnaud, 1801, can inflict significant damage to canola crops before the four-leaf stage leading to plant death (Nash, Thomson & Hoffmann, 2007). The increased pest status of slugs is often attributed to the retention of crop residues which serve as habitat and food (Glen, 1989). It may also be due to a reduction in key predator numbers (e.g., predatory beetles Carabidae and Staphylinidae) or a change in broader predator communities (including spiders, ladybeetles, lacewings, predatory mites) as a result of widespread pesticide use (Nash, Thomson & Hoffmann, 2008a), including insecticidal seed treatments (Douglas, Rohr & Tooker, 2015).

To predict the impact of pesticides on the interaction between pest species abundance, natural enemy abundance, and crop yield we analyse the change in ground-dwelling invertebrate community composition under application of organophosphates and across a rotation sequence in a commercial grain field. We first investigate how the prophylactic use of pesticides to control earth mites at the establishment stage of crops impacts both pest and natural enemy invertebrate communities. Secondly, we investigate whether structural change to the natural enemy invertebrate community over a period of three seasons led to the outbreak of secondary pest species, in this case slugs. We make an assessment of the yield effects that may be attributed to the trade-offs involved in pesticide applications and discuss both how our data support IOBC ratings, and how growers could use this information in decision-making.

Materials & Methods

Study site and experimental design

The study site was situated near Mortlake, Victoria, Australia (38°00.5′S, 142°45.3′E), which has a temperate climate with mean maximum annual temperature of 18–21 °C and mean annual precipitation is 625 mm. The soils are predominantly of grey sodosols (Isbell, 1996) based on quaternary basalt. The experimental area was located in a 36 ha field managed according to standard district agricultural practices using 2 m wide raised (20 cm) beds, constructed to alleviate water logging. To limit confounding non-target impacts from other chemicals, no seed treatments or fungicides were used during this trial, however herbicides were applied. The field was divided into 72 m wide strips and each was allocated to one of three treatments; methidathion (Supracide®; Syngenta, Basel, Switzerland) representing conventional district practice, chlorpyrifos (Lorsban®; Dow AgroScience, Zionsville, IN, USA) representing a supposedly more disruptive treatment, and 80 l/ha of water as a control (Fig. 1). The dates of sowing and treatment applications, and the seasonal weather conditions for each year are shown in Table 1. In 2004 and 2005 the pesticides were applied at recommended field rates of 40g active ingredient (a.i.) /ha for methidathion and 250 g a.i./ha for chlorpyrifos. Field observations indicated that treatments did not control pests in 2005–2006, so the rate of chlorpyrifos was increased to 500 g a.i. /ha for 2006 and 2007. This double rate was selected as it (i) reflected grower practice when responding to a multitude of establishment pests, and (ii) may increase disruption to natural enemy communities through an increase in the magnitude of exposure to a known toxicant (Table 2). This provided an assessment of the effect of increased application rates on the pest and natural enemy community and is similar to real-world practices where pesticide spray rates are increased in response to visible signs of high pest abundance, and/or damage.

Figure 1 Site layout.

Site layout indicating insecticide treatments and invertebrate sampling transects (blocks) in relation to overall yield from the 36 ha field (pooled data from 2003–2007).

Table 1 Sowing dates, seasonal and experimental conditions throughout the study period at the experimental site near Mortlake in Victoria.

	Crop sown	Planting date (treatments applied)	Growing season rainfall (mm)	Average yield in region t/ha	
2004	Barley (Gardner)	*15 Oct 2004
(19 Oct)
	Sep–Feb 310	Wheat 2.06
Canola 1.05	
2005	Wheat (Kellalac)	28 Jun 2005
(1 Jul)	Apr–Nov 348	Wheat 2.67
Canola 2.31	
2006	Wheat (Kellalac)	17 Jun 2006
(21 Jun)	Apr–Nov 269	Wheat 2.75
Canola 1.7	
2007	Canola (Thunder TT Pacific seeds)	5 June 2007
(6 Jun)	Apr–Nov 470	Wheat 5.37
Canola 2.65	
Notes.

* Very delayed sowing this season due to seasonal conditions. The yield for Kellalac from Hamilton National Variety Trials for each season is included, as is the site mean for Triazene Tolerant canola varieties for comparative purposes as obtained from relevant trial reports (http://www.farmtrials.com.au/trial_report_library.php?action=search&query=Hamilton accessed 13 Feb 2017).

Table 2 Natural enemy and pest communities and taxa defined for this study.

IOBC (International Organization for Biological and Integrated Control Pesticides and Beneficial Organisms) ratings for representative members of each beneficial group are included, as well as which target pests are registered for the respective chemical use in Australia. An empty cell indicates no available information. IOBC ratings for the natural enemies are taken for a representative from that grouping. IOBC toxicity ratings are on a 4 point scale (1 “harmless” <25% mortality to 4 “very harmful” >75%). The registration for pests is taken from the chemical labels for the respective pesticides (Lorsban (chlorpyrifos), Dow Chemicals, APVMA Approval No: 32887/56655; Supracide (methidathion), Syngenta, APVMA Approval No: 33041/5). Registration is marked (Y)es if the target pest is included for grains crops, if there is registration for a target pest outside of grains, it is included in parentheses.

Natural enemies	
Group	Includes	IOBC— Chlorpyrifos	IOBC— Methidathion	Total	
lacewings	Micromus tasmaniae	4	4	61	
carabids	Coleoptera: Carabidae	2–4		413	
staphylinids	Coleoptera: Staphylinidae	3–4		414	
ladybirds	Coleoptera: Coccinellidae	3–4		14	
wasps	all hymenopteran parasitoids	1&4		144	
predatory bugs	predatory bugs, e.g., assassin bugs	4	3	96	
spider1	hunting spiders (Lycosidae, Miturgidae, Lamponidae)	3		493	
spider2	web-building spiders and harvestmen (Linyphiidae, Opiliones)		3	182	
spider3	sac-spiders (Clubionidae, Gnaphosidae, Corinnidae)			96	
salticids	jumping spiders (Salticidae)			18	
predatory mites	snout mites, mesotig mites, trombididae mites	4		225	
syrphids	hoverflies			70	
centipedes	all centipedes			7	
Pests	
Group	Includes	Registered—Chlorpyrifos	Registered—Methidathion		
slaters	Isopoda: Armadillidiidae			1,745	
millipedes	Ommatoiulus moreleti (Portuguese millipedes)			960	
slugs	Milax gagates, Deroceras reticulatum			305	
earwigs	predominantly Forficula auricula, some natives	(Stonefruit)		1,280	
earth mites	Halotydeus destructor, Penthaleus spp., Balaustium spp.	Y	Y	159	
aphids	Rhopalosiphum padi, Myzus persicae	Y	(Lucerne, Lupins etc.)	3,349	
lucerne flea	Sminthurus viridis (Collembola: Sminthuridae)		(Lucerne)	23	
scarab beetles	Coleoptera: Scarabidae	Y	(Pasture)	165	
lepidopterans	pest catepillars	Y		82	
weevils	Coleoptera: Curculonidae	Y	(Lucerne)	556	
orthopterans	crickets and grasshoppers	Y	(Ornamentals)	625	

Invertebrate sampling

Since 2005, three transects (henceforth referred to as blocks) were laid out perpendicular to the treatment strips, so that each bisected the three treatments, achieved maximum interspersion of treatments and achieved spatial independence (Fig. 1). The edges of the strips were avoided by sampling towards the middle of the 72 m wide strip. In each block, five census points (12 m apart) were selected per replicate (total of 5 × 3 × 3 census points). Each census point consisted of four pitfall traps (7 cm deep by 11 cm in diameter filled 190 ml ethylene glycol) to capture macro-invertebrates (>2 mm), and four surface refuge traps to capture slugs (300 mm by 300 mm terracotta paving tile placed on the soil surface as per Nash, Thomson & Hoffmann, 2007). While these sampling methods are not optimal for all macro-invertebrates in the field, it does allow for an assessment of the ground-active communities.

Sampling was conducted three times a year to coincide with crop establishment (Zadok’s Growth Stage 1) (herein referred to as winter), stem elongation (Zadok’s Growth Stage 3) herein referred to as spring) and post-harvest (herein referred to as summer) (Table 1). Traps at each census point were established after sowing and opened for one week, before being closed until later sampling when reopened for one week. The pitfall trap catch was returned to the laboratory and sieved (200 µm mesh) prior to sorting under a dissection microscope. The refuge traps were turned over in the field and the number and identity of the slugs on the underside recorded in the field, then all individuals were removed with a subset retained in 70% ethanol as vouchers.

From the invertebrate data collected in the pitfall traps we defined two functional communities, pests and natural enemies, and examined changes within these two broad groups as well as targeted analysis on individual species or taxa (Table 2). As some taxa such as millipedes and earwigs may sometimes act as either pest or natural enemy it can be difficult to broadly classify them at this level. Here we assigned them as pests as they may inflict damage, but may also act as natural enemies at certain times. Taxa such as ants (Formicidae) are also hard to assign to one of these two groups in this system, as they perform roles outside of pests and natural enemies, and so were omitted from subsequent analysis. For some common pest and natural enemy species, their identification is straightforward. However, for many taxa we lack species descriptions (especially for immature stages) and in these cases family-level identification was conducted.

Yield

Yield of the crops grown during this study were recorded using Advanced Farming Systems (AFS) features available for Case IH combine harvester, with different machines being used over the course of the study. The software recorded clean grain flow and moisture whilst harvesting, along with geographic co-ordinates (WGS 84). The data was calibrated in accordance with individual user manuals, but to ensure accuracy actual weights obtained from point of sale were used to correct data to tonnes/ha for each season. Initial data handling and maps were developed using the SMS™ Advanced Software Ver. 8.0 (Ag Leader Technology®, Ames, IA, USA). Spatial yield data was gridded using kriging with the default settings (linear variogram) in Surfer©Ver. 8.05 (Golden Software, Inc. Colorado), to create contour maps to compare with the invertebrate samples collected in the transects. Because of differences in collection of spatial data, tolerances were set at 10 m2 for yield data, and 6 m2 for invertebrate census points, not all points overlayed exactly so corresponding data was matched and extracted manually. Geographic referenced information was converted to Cartesian coordinates using the software GEOD Ver. 3.42 (Graham Samuel and Associates Pty Ltd, Charlestown, NSW).

Statistical analyses

Due to the strip treatments being inherently wide (resulting from the equipment used), it was not possible to randomise the treatments within a block, making spatial non-independance a potential issue. Prior to the final choice of methods, we conducted some preliminary analysis of our data using a very conservative approach (generalised estimating equations (GEE); Dormann et al., 2007). In this approach, edge groups of 4 traps were included and, or, excluded to check for migration and compensate for spatial non-independence. We found that data did not violate assumptions (GEE run with and without an autoregressive function for position within treatment strip gave the same conclusion) and migration between treatment strips was not detected. These preliminary analyses supported that populations here had a random distribution, and independence, and that there was not a requirement to randomise the treatments.

We calculated species richness as total number of taxa present (prior to defining pests and natural enemy communities), for each sample across treatments and sampling times to examine overall effects of pesticide application across the study period. We assumed that each taxa represented one species (even if sorted to family-level), although this is likely to underestimate species diversity. We calculated species turnover within each treatment through time, using the “codyn” package in R (Hallett et al., 2016). This analysis allows for the total turnover to be calculated per time point (i.e., the proportion of species that differ between time points either by appearing or disappearing), but also the proportion of species that appear and disappear at each time point. We then examined the mean rank shifts for each treatment, which give an indication of the degree of species reordering between two time points, again using the codyn package (Hallett et al., 2016).

To examine the effects of pesticide on both pest and natural enemy communities, we employed two different methods that have recently been evaluated for use in ecotoxicology (Szöcs et al., 2015). The first of these are principle response curves (PRC; Van den Brink & Ter Braak, 1999), which are widely used in freshwater mesocosm studies and pest-resistant crops such as Bt cotton, to examine community level response to pesticides (or control strategies) over time (e.g., Naranjo, 2005; Whitehouse, Wilson & Fitt, 2005; Rose & Dively, 2007). A PRC is a time-dependent multivariate technique based on RDA (redundancy analysis). By incorporating treatment, time and the interaction thereof, a PRC allows for visualizations of a treatment effect through time on community structure, by highlighting variance in overall response. We used the function “prc” in the R package “vegan” to conduct these analyses. To evaluate the significance of treatment at each sampling point, we conducted single RDAs and used a permutation structure to account for the blocked design of the study.

The second method fits separate generalised linear models (GLMs) to each species, to give an overall community analysis (Wang et al., 2012) and has recently been applied to investigate pesticide effects on communities (Szöcs et al., 2015). By using Poisson or negative binomial distributions, GLMs are able to handle to count data without the need for data transformation. Methods that incorporate GLMs appear to be particularly suited for identifying responding taxa that would be missing on the first axis in the PRC (Szöcs et al., 2015). Principal response curves appear to describe the direction of the effect on the community more clearly, thus using PRC and GLMs together allows for complimentary analysis on community and treatment data through time. Using the R package “mvabund” and function “manyglm” we ran separate GLM per species, using negative binomial distributions, for the two communities (pests and natural enemies), with the three treatments and time points, and the interaction thereof, as the dependent variables. We compared models incorporating treatment and time to models without to investigate the overall effect of treatment, and to investigate the interaction of treatment and time. Finally, we performed separate analyses at each sampling time point to examine differences between communities for each treatment, using Likelihood Ratio tests on the univariate responses (species) and 2,000 bootstrap repeats. To visualise changes in the pest and natural enemy communities in response to the treatments, described through the GLM analyses, we plotted the combined deviance (effect size) for the members of each community across each of the sampling points.

Assessing the ultimate impact of the pesticide treatments on crop yield was challenging due to the large seasonal fluctuations in conditions for crop growth and underlying spatial patterns in crop productivity across large fields. However, ideally every pesticide input should provide some yield benefit for the grower each year, regardless of seasonal conditions, usually through protecting the crop from damage due to pests. In our study this should manifest as a significant increase in crop yield in the treated parts of the field in comparison to the control area. Prior to analysis of yield we removed outliers (high values for yield) that corresponded to instances where the header stops during harvest and extra ingrain is collected. These outliers were identified as being further than two standard deviations from the mean, per block, per year. We then performed separate season GLMs with the yield as response, and treatment and block as fixed effects, including the interaction between treatment and block. As there was almost always a significant interaction, pairwise contrasts using the “lsmeans” package in R, were used to determine where treatments differed within blocks.

Results

In total 115 different species or taxa were identified from our samples: This included 10 species of carabid mostly belonging to the subfamilies Broscinae and Pterostichinae and including a key slug predator, Notonomus gravis Chaudoir, 1865; seven species of staphylinids, and two species of slug (see Table 2 for broader classifications). Prior to the increased rate of chlorpyrifos application, the initial winter samples at crop emergence in July 2005 were the lowest in species richness, this contrasts to the October 2005 spring sample yielding the highest species richness. This large amount of seasonal variation in community composition is further highlighted by the first 2006 sample yielding low species richness again (Fig. 2A). Importantly richness through time suggests there was no difference between treatments for the first three sampling points, supporting the rationale behind increasing the chlorpyrifos application rate. Species turnover was high and similar across treatments (65–80%) for the samples prior to the rate increase of chlorpyrifos, again reflecting the seasonal nature of the species examined (Fig. 2B). After the chlorpyrifos concentration was increased in 2006, species richness diverged between treatments (Fig. 2A), with chlorpyrifos having the highest richness in the spring 2006 sampling, before the lowest in the winter 2007 sample. The methidathion treatment had a higher richness than the control and chlorpyrifos in the last two sampling time points. Species turnover began to differ between treatments following the increase in chlorpyrifos, and is much more variable in the chlorpyrifos treatment than the control or methidathion. Over the course of the entire experiment, the mean rank shift pattern reflects richness and turnover, and suggests that variability between treatments for the abundance of different species becomes increased through time, compared to the control (Fig. 2C).

Figure 2 Species richness.

(A) Change in total species richness over time, total of 115 taxa. The grey bars represent the sampling times and the red dashed lines represent the application of pesticides associated with each treatment. (B) Total proportional species turnover for each time point through the study period. (C) Mean rank shifts. Note for (B) and (C) the initial sampling and spray event is not present, as each point represents the change from the previous sampling event. The first spray event is immediately before the beginning of these panels, however.

To display how key species from the PRC and multiple GLM analyses (see below) changed in abundance patterns through the trial, we plotted the temporal abundance per treatment for slugs and their potential predators, carabids and staphylinids, as well as other species displaying large responses: earth mites and earwigs (Fig. 3). Outside the summer samples in 2006 and 2007 (reflecting their seasonality due to lack of moisture post crop harvest), slugs were consistently more abundant in the chlorpyrifos treatment than the control and methidathion. This is in contrast to other pest species such as earth mites and earwigs, which display lower abundance in the organophosphate treatments, especially towards the end of the trial (Fig. 3). While some of the slug abundance patterns may be explained by less mortality from predators, the response of the main predator groups (in this dataset) is complex. The carabids initially show quite high abundance, but then for July 2006 and June 2007, carabid numbers in the chlorpyrifos treatment are well below the control and methidathion treatments. After the 2006 pesticide application (including the higher rate) carabids are absent in the chlorpyrifos treatment, though there are only a few individuals in the methidathion and control treatments. Carabid numbers recover and increase from this point, possibly responding to the high abundance of slugs, until the 2007 pesticide application: following this event the carabids are reduced again to zero in the chlorpyrifos treatment, whilst persisting in the methidathion and the control treatments. The staphylinids were heavily reduced in numbers in the chlorpyrifos treatment (and methidathion but not to as great extent) following the 2006 pesticide application.

Figure 3 Selected species abundances.

Abundances (untransformed count data) through time for selected pests and natural enemies displaying important responses in the analyses (see Figs. 4–6 and Table 3). Pests (A) slugs, (B) earth mites and (C) earwigs, and the predatory beetles: (D) carabids and (E) staphylinids.

The natural enemy community initially displayed an increase over the control as shown by the principal response curve (Fig. 4A). In 2006 the effect switches to become negative, and for methidathion it stays negative. Chlorpyrifos, however, goes back to a positive effect at the final time point. The carabids are strongly weighted against the community trend, indicating that they likely had fewer numbers in the chlorpyrifos treatment by the end of the study period. Predatory bugs also do not follow the treatment effects on the community patterns. For the pests PRC (Fig. 4B), there was no differences between the control and chlorpyrifos or methidathion for the initial applications. Target pests (earwigs, earth mites and millipedes) all exhibit strong positive weightings to the negative effect of the pesticide applications, in particular chlorpyrifos for earwigs and earth mites, methidathion for millipedes (Fig. 4B). Unlike the target pests, slugs show an opposite trend towards the temporal pest community response (Fig. 4B).

The multiple GLM approach broadly agrees with the results from the PRC, as reflected in significant (P < 0.05) and non-significant (P > 0.05) community differences at most of the same sampling periods. The exception is that for the pest community, the GLM approach determined the July 2005 and September 2006 samplings as significantly different from the control, with the PRC not significant at the 0.05 level (0.077 and 0.087, respectively). For both the natural enemies and pests, the community was significantly affected following the increase in the rate of chlorpyrifos (After July 2006, Table 3). The carabids and staphylinids showed the largest contributions to the overall community trends, with 23.7 and 13.4%, respectively (Fig. 5C). The carabids were significantly affected in September 2006 and June 2007 (Table 3), whereas the staphylinids in July 2006 (Table 3). Other species, such as the predatory bugs (June 2007) and ground dwelling wasps (July 2006) also display significant effect sizes following the spray events (Fig. 5C; Table 3). The multiple GLMs on the pest community indicates that effect sizes were also greatest following the spray events (Fig. 6A), especially for chlorpyrifos (Fig. 6B). Overall, the pest community in all but two samples (both prior to increased rate of chlorpyrifos) was significantly affected by pesticide application (Table 3). The effect sizes also appear to increase with time, but this may also be due to the final crop in the rotation being canola, where mites, aphids and earwigs were all significantly affected by the organophosphate treatments. The earwigs and isopods (slaters) had the largest contributions across the trial (16.5 and 12.2%, respectively; Fig. 6C), but this also appears mostly to be due to the last sample being in the canola phase. Interestingly, the abundance of isopods (slaters) was higher in the chlorpyrifos treatment for the final sample. Weevils, millipedes and orthopterans also provided contributions between 10–11% each (Fig. 6C). While most of these pests displayed negative abundances and larger effect sizes following a spray event, the effect size for the slugs is greatest in September 2006, when there was high abundance in the chlorpyrifos.

Figure 4 Principal response curves.

(A) Principal response curve for the natural enemy community. The left y-axis shows the Effect size. The position on the right y-axis reflects the weighting of the species to the overall response. The 0 line reflects the untreated control. (B) Principal response curve for the pest community. The left y-axis shows the Effect size. The position on the right y-axis reflects the weighting of the species to the overall response. The 0 line reflects the untreated control.

Figure 5 Beneficial community.

Effect size through time on the natural enemy community identified in this study. Each species grouping has had individual generalised linear models performed on abundance. The effect size is relative to the control, and the different colours represent the species contribution to that effect size, at that point in time. The dashed vertical lines represent the application of the pesticide treatments. (B) The overall community response to the application of the treatments through time. For both (A) and (B), the control is represented by the 0 line, and the treatments correspond to the deviance from the control. The three dashed vertical lines represent the application of the pesticide treatments. (C) The proportional contribution of each species to the overall deviance.

Figure 6 Pest community.

Effect size through time on the pest community identified in this study. (A) Each species grouping has had individual generalised linear models performed on abundance. The effect size is relative to the control, and the different colours represent the species contribution to that effect size, at that point in time. The three dashed vertical lines represent the application of the pesticide treatments. (B) The overall community response to the application of the treatments through time. The control is represented by the 0 line, and the treatments correspond to the deviance from the control. The four dashed vertical lines represent the application of the pesticide treatments. (C) The proportional contribution of each species to the overall deviance.

Table 3 Significance of treatments from Principal Response Curve (PRC) and multiple GLM analyses for the beneficial and pest communities, at each sampling time.

The PRC analysis score is for the whole community and reflects the redundancy analysis (RDA) score for that single time point. The GLM scores are for significance of the community or species deviance related to the treatments at each sampling time. All bold values indicate significant score (p < 0.05).

Beneficial community	
		Community	Groups	
Year	Month	RDA (PRC)	GLM	lacewings	wasps	carabids	staphylinids	predatory mites	ladybirds	syrphids	centipedes	spider1	spider2	spider3	salticids	predatory bugs	
2005	July	0.571	0.442	1.000	1.000	0.937	0.639	1.000	1.000	1.000	1.000	0.639	1.000	1.000	0.639	1.000	
	October	0.697	0.515	0.930	0.866	0.975	0.991	0.991	0.796	0.929	0.983	0.991	0.974	0.983	0.991	0.783	
2006	March	0.695	0.312	1.000	0.997	0.997	0.068	0.997	1.000	1.000	0.633	0.733	0.997	1.000	1.000	0.984	
	July	0.001	0.000	1.000	0.009	0.088	0.038	0.021	1.000	1.000	0.547	0.547	1.000	1.000	1.000	1.000	
	October	0.001	0.002	0.057	0.648	0.042	0.195	1.000	0.648	0.271	0.648	0.648	0.648	0.648	0.648	0.951	
2007	March	0.272	0.374	1.000	0.818	0.865	0.818	0.721	1.000	1.000	1.000	0.865	0.818	0.721	1.000	0.865	
	June	0.001	0.000	1.000	0.732	0.001	0.933	0.933	1.000	1.000	0.885	0.933	0.227	0.826	1.000	0.019	
Pest community	
		Community	Groups	
Year	Month	RDA (PRC)	GLM	slaters	millipedes	slugs	earth mites	lucerne flea	aphids	earwigs	scarab beetles	lepidopterans	weevils	orthopterans	
2005	July	0.067	0.028	0.230	0.230	0.968	1.000	1.000	1.000	0.968	0.069	1.000	0.656	0.643	
	October	0.593	0.116	0.733	0.707	0.579	1.000	0.274	0.948	0.974	0.961	0.809	0.961	0.529	
2006	March	0.859	0.713	0.946	0.976	1.000	1.000	1.000	1.000	0.976	0.976	0.976	0.976	0.577	
	July	0.028	0.025	0.446	0.500	0.446	0.579	1.000	1.000	0.108	0.749	1.000	0.500	0.617	
	October	0.087	0.011	0.495	0.421	0.038	1.000	1.000	1.000	0.116	0.221	1.000	1.000	0.274	
2007	March	0.003	0.006	0.346	0.346	1.000	1.000	1.000	1.000	0.555	0.555	0.656	0.007	0.285	
	June	0.004	0.000	0.017	0.703	0.703	0.018	1.000	0.027	0.005	1.000	0.059	0.426	0.703	

There was a large amount of spatial variation in the yield in the areas of the field corresponding to the difference treatments, and this was further complicated due to an interaction with the block (Fig. 7). For the barley (2004 crop) yield, harvested in 2005, there was an overall treatment effect (χ22=7.15, p < 0.03), but this appears due to block 3 (yield in the Control was significantly lower than yield in both treatments). Following this, there was a significant block effect (χ22=10.44, p < 0.001) and significant interaction between the treatments and the block (χ42=14.32, p < 0.001) (Fig. 7). Following an increase in the rate of chlorpyrifos, for the wheat (2005 crop) yield harvested in 2006 the overall treatment effect was significant (χ22=43.38, p < 0.001) with yield significantly higher in the chlorpyrifos treatment than the control in blocks 1 and 3, but lower in block 2 (all significant, p < 0.001). Methidathion yielded significantly lower than the control and chlorpyrifos in blocks 1 and 3 (p < 0.05), and significantly higher than chlorpyrifos in block 2 (p < 0.001). These differences gave an overall significant block effect (χ22=96.99, p < 0.001) and a significant overall interaction effect (χ42=162.90, p < 0.001) (Fig. 7). For the wheat (2006 crop) harvested in 2007, the chlorpyrifos gave consistently higher yields than the control and methidathion in blocks 1 and 3 (p < 0.01), and there was a significant overall Treatment effect (χ22=65.19, p < 0.001). There was no significant difference between all treatments in block 2. The control and methidathion were only significantly different from one another in block 1. Again, there was a significant block effect (χ22=37.10, p < 0.001) and a significant overall interaction (χ42=40.99, p < 0.001) (Fig. 7). Finally, for the canola yield harvested in 2008, there was no data in blocks 1 and 2 as seedlings were completely lost due to slug herbivory at establishment (July 2007), regardless of the treatments (i.e., the treated areas still suffered the same damage). In the remaining block (3), chlorpyrifos (at the higher application rate) was significantly lower in yield than the control and methidathion treatment (p < 0.001) and there was an overall significant treatment effect (χ22=56.91, p < 0.001) (Fig. 7).

Figure 7 Yield.

Yield (tonnes per hectare) per crop type and season, treatment and experimental transects (blocks 1–3).

Discussion

Although the overall interactions of season, pesticide application and crop type on both pest and natural enemy communities are complex (e.g., Brust, Stinner & McCartney, 1985; Holland & Luff, 2000), this study provides some indication of how crop rotation can interact with pesticide use. Importantly, community composition was very similar to the control with standard rates of pesticides, however after higher rates were used, the change in community composition was marked. We demonstrated that such pesticide applications are likely to come with trade-offs associated with the reduction in important generalist predatory species, and that the timing of these reductions may have profound effects on pest suppression and crop production.

Not all invertebrates will be directly affected by organophosphates in the field, but disruption of important predators at critical times (e.g., a certain crop type) may be more consequential to pest suppression than overall community effects. The final rotation into canola here demonstrates how the reduction of carabids at this point was more detrimental to the grower than the reductions of either carabids and staphylinids in the prior wheat rotations, and the subsequent outbreak of slugs in the increased chlorpyrifos treatment appears indicative of a secondary outbreak. Whilst it is difficult to draw a causal link between absence of predators and the outbreak of slugs in this study, the reduction of slugs by carabids has been demonstrated in similar systems (Nash et al., 2008). Our results provide an interesting contrast to a recent study that reported that imidacloprid applications increased slug issues due to disruption of adequate biological control as non-target effects (Douglas, Rohr & Tooker, 2015). Such field response data are important, as there are few studies that use field evaluation of non-target effects of pesticides (e.g., Stäubli et al., 1984; Thomson & Hoffmann, 2006; Jenkins et al., 2013), most studies typically use laboratory bioassays, or short-term small-plot trials (e.g., Macfadyen & Zalucki, 2012; Macfadyen et al., 2014). Further testing of acute and sub-lethal effects under semi-field conditions is required to test our findings, and like Jenkins et al. (2013) we suggest that laboratory assessments of toxicity should be extrapolated with caution to the field setting.

In Australia, short-term semi-field studies (Jenkins et al., 2013) have suggested that the impacts of chlorpyrifos are not as disruptive to natural enemies as previously thought (Curtis & Horne, 1995), however cumulative impacts over longer time periods are considered disruptive in viticulture (Nash, Hoffmann & Thomson, 2010) and arable systems (Nash, Thomson & Hoffmann, 2008). The strong negative response of carabids (Pterostichinae) to chlorpyrifos is concordant with overseas data on the closely related Pterostichus melinarius, with slightly higher rates (720 g a.i.) being considered harmful (IOBC rating 3) (Hassan et al., 1988). Lower rates (480 a.i.) have been shown to be less harmful (IOBC 2) to the carabid Bembidion sp, in field trials when compared to lab assays (Floate et al., 1989), however toxicity responses vary between studies and methods, ranging from IOBC 2–4 (Cockfield & Potter, 1983; Bale, Ekebuisi & Wright, 1992; Turner, Bale & Clements, 1990). There is limited data on methidathion impacts on natural enemies, however it has been considered as very harmful (IOBC 4) to green lacewings in semi-field trials (Hassan et al., 1985) and harmful (IOBC 3) to spiders, predatory bugs and green lacewings in the field (Stäubli et al., 1984). We did not find methidathion to cause significant reductions on those taxa, however this may be due to the lower rate applied here (40 g a.i.) compared to a previous field study (120 g a.i.) (Stäubli et al., 1984). The lack of significant population reductions may also be due tothe sampling methods used do not optimally target these natural enemies, or perhaps there was some avoidance of the winter pesticide application due to their activity at later crop stages. Some species may also have higher levels of tolerance to organophosphates, making results for those taxa that were not strongly impacted by pesticides, as relevant as for those that were.

Chemical control is the dominant control option for major pests such as earth mites. While target pests in this study were all controlled, there was no discernible yield response warranting application targeting these pests. Within the common earth mite species there are varying susceptibilities to organophosphates, including methidathion: Penthaleus falcatus has a higher tolerance to methidathion than either of P. major, P. tectus or Halotydeus destructor (Umina & Hoffmann, 1999). Earwigs (including Forficula auricula) are widespread in southern Australian grain systems, and although they are typically considered as sporadic pests (e.g., Murray, Clarke & Ronning, 2013), their role as a pest or beneficial species is presently unclear. In addition to acute toxicity and high rates of mortality, chlorpyrifos-ethyl has been shown to reduce the predatory behaviour of the earwig F. auricula in orchards (Malagnoux, Capowiez & Rault, 2015), where they are considered effective biological control agents. The strong response of earwigs to organophosphates suggests that any form of pest suppression service in grains crops could be hampered by the application of harmful organophosphates such as chlorpyrifos. Despite some limited data suggesting moribundity (Kassebaum, 1985), there is little known about the ability of organophosphates to control millipedes, and there are no currently registered chemicals for control in Australian grains.

Undertaking this study over four seasons and different crop rotations gives some indication of the trade-offs and long-term effects of pesticides on the pest species response, and the response of the natural enemy communities that co-occur with them. Most of the key invertebrate species here would have undergone multiple generations during the experiment, which implies that some sublethal effects (e.g., reduced fecundity, survivorship of immature stages, short-term toxicity effects) should have been captured during this experiment. While changing the application rate of chlorpyrifos during the experiment is not ideal from an analysis point of view, it provides a realistic scenario of how growers adjust rates and frequency of application in response to perceived pest threats. The continued use of organophosphates as a default for control of insect pests in Australian grains is perhaps best exemplified by an emergency permit for use of chlorpyrifos and Pirimicarb (APVMA 82792) to control Russian Wheat Aphid (Diuraphis noxia Kurdjumov 1913) in response to its incursion in 2016 (Yazdani et al., 2017).

We did not observe spatially consistent yield benefits from applying pesticides, and in canola the application of pesticides did not prevent widespread seedling loss from slugs (in a season with good levels of rainfall for slug populations, Table 1). Furthermore, there was only one instance where the control had significantly lower yield than both the chlorpyrifos and methidathion treatments, and this benefit was not seen across the whole study area (barley 04/05 block 3, Fig. 7). Large spatial variation within the field in the yield response may be related to spatial variation in the abundance of pests and natural enemies. However, given the significant interactions between pesticide treatments and blocks used to control for spatial variation, it is likely that other invertebrate species that we did not record may be involved. The detrimental effects of chlorpyrifos on the key slug predators may be evident from the lower yield seen in the last remaining block of canola during the final year of the study. Unfortunately, having only one block surviving makes it hard to test this pattern further.

The lack of a spatially consistent yield benefit from the application of organophosphates suggests that growers could limit broad-spectrum pesticide applications without risking any crop losses due to invertebrate pests. This could be achieved through either applying a threshold-based approach to spray decisions, or selectively targeting areas of the field that may be at risk. For H. destructor a recently published study examined thresholds associated with economic crop losses (Arthur, Hoffmann & Umina, 2015), and recommendations for control have called for rotation in the use of chemicals, non-chemical management options and crop rotations (National Insecticide Resistance Management Working Group, 2016). Managing for control failures due to resistance in pests is an important component of grains pest management in Australia, and the risk of secondary outbreaks requires similar attention. As much of Australian grains production includes rotation with other crops, understanding and responding to the risk of secondary pest outbreaks will require growers to manage their pesticide-use across an entire rotation.

Conclusions

Demonstrating the long-term effects of organophosphates on the ecology of invertebrate species within Australian grains systems is complex, due to the scale of production, diverse rotation practices, and inter-annual variation of species diversity and abundance. Despite this, there are important points that arise from this experiment conducted across a standard crop rotation. Firstly, the prophylactic use of organophosphates as a management strategy requires understanding of the risks of secondary outbreak in both the current crop and subsequent crops in the rotation. Secondly, quantifying the impact of reactive management strategies (such as increasing pesticide rates) on pest and natural enemy communities will allow growers to make more informed judgements on the risk of disrupting biological control services.

Eduard Szöcs provided much of the code for the multiple GLM analysis on his blog (https://edild.github.io). Lewis Wilson provided useful comments on a previous version of this manuscript.

Additional Information and Declarations

Competing Interests

Author Contributions

Data Availability

The authors declare there are no competing interests.

Matthew P. Hill analyzed the data, wrote the paper, prepared figures and/or tables, reviewed drafts of the paper.

Sarina Macfadyen contributed reagents/materials/analysis tools, wrote the paper, reviewed drafts of the paper.

Michael A. Nash conceived and designed the experiments, performed the experiments, contributed reagents/materials/analysis tools, wrote the paper, prepared figures and/or tables, reviewed drafts of the paper.

The following information was supplied regarding data availability:

Hill, Matt; Macfadyen, Sarina; Nash, Michael (2017): Broad spectrum pesticide application alters natural enemy communities and may facilitate secondary pest outbreaks. figshare.

https://dx.doi.org/10.6084/m9.figshare.5263462.v3.

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
