# Peer review of "Broad spectrum pesticide application alters natural enemy communities and may facilitate secondary pest outbreaks"

_PeerJ, doi:10.7717/peerj.4179_

## Round 0.1 · original submission · Major Revisions

I have received two set of reviews for your paper. Both reviewers point to shortcomings in the data analysis. Reviewer 1 also suggests cutting down the overall length of the paper. I request you address these comments and resubmit your paper after major revisions. Please see the comments below.

Reviewer 1 ·

Basic reporting

The article is very readable, though too long.
Raw data is not shared. Don't know if this is mandatory.
Not an expert in terrestrial ecotoxicology, cannot comment on whether literature background is exhaustive.

Experimental design

Design largely sound but has shortcomings, see below

Validity of the findings

Discussion too long.

Additional comments

The paper describes the effects of two pesticide treatments on pest and beneficiary invertebrates over several seasons. The study design is largely sound, but see major comment below. The article is very readable, but pretty long given its contents.

Major comment
The data analysis seems flawed to me. The authors seem to treat the replicates within a block as independent. However, they are not and the block design does not help here. Generally, there are two replicates per treatment, thus inferential statistics seem misplaced to me. At least in aquatic mesocosm studies this would not work, and I don't see why sampling the same treatment strip multiple times would result in treatment replication.
Moreover, the sample size would be even smaller if factors such as the crop type or amount of pesticide applied would be included in the analysis (I don't suggest to do this, but the sample size is too small to draw general treatment conclusions based on statistical inference).

Minor comments
19 "community ecology" specify, what you examine
29f point out what is pest and beneficiary
35 add whitespace 36ha
60 and but
92 Begin sentence with capital letter
138 replace comma with .
187 should be "were"
189 how was the kriging done? Should also be kriged data
209 delete "studies"
235 what kind of null model?
257 suggest to move "in total" to beginning of sentence
218 suggest to rephrase "this contrasts to"
308 whitespace missing (pest community)
391 replace methodologies with methods
395 remove "()"
406 species (spelling)
Figure 1 you could put the scale directly beside the figure so as to clarify that each width of treatment is 78 m
Figure 4 Spelling should be "shows" instead "dhows"
Figure 5b control could be included in plot

·

Basic reporting

No comment

Experimental design

No comment

Validity of the findings

No comment

Additional comments

This study represents an impressive amount of sampling and sorting to address a question of great interest and which requires considerable commitment to address with field studies. Represents an impressive attempt on the part of the authors to identify the factors affecting pest and natural enemy abundances in a field study.

It would be useful to be provided with information as to relative abundances of taxa considered eg the importance of the observed a significant impact on predatory beetles (carabids and staphylinids) is related to their abundance.
See also Ll310 Ff clearly abundance of these taxa is relevant to effect size also L315 with reference to predatory bugs and wasps. This information could be inserted to Table 2.

The authors indicate they aim to sample ground dwelling invertebrates but include several beneficials (eg lacewings, ladybirds, syphids) and even pests (eg aphids) for which this is a suboptimal sampling method. Some mention of this should be made.

Include consideration of the role of increased rainfall and change of crop to canola in increase in slug abundance observed. Is there potential for increased slug abundance in the final sampling be influenced by the increased rainfall in that season?

Natural enemy and beneficial are used interchangeably throughout ms. Greater consistency should be aimed at especially in referring to results where tables and figs use beneficial rather than natural enemy.

L84 delete c
L85 suggests
L135-137 check font

Ll138-139 Meaning unclear-“Seed treatments and fungicides were not applied during this long term experiment, only herbicides to limit confounding non target impacts from other pesticides”. Re write to clarify

L149 L why was double application of chlorpyrifos expected to increase impact on beneficials? Reference or justification required

Ll152-153 repeat of LL143-144. Please delete

L173 add information clarifying the outcome of sorting and grouping eg resulting in 13 taxa (variously order, family, species) classified as “beneficials” and 11 taxa of “pests” (see Table 2 for detail). And use consistent term in table “taxa” rather than groups

L198 species richness-all taxa? Both “pest” and “beneficial”

In results need to list the taxa as included in analyses

Table 2 “Beneficial and pest communities and groups defined for this study”


These are the individual Taxa referred to in the text (L) I presume. This needs to be made clear and the same terminology used. Change caption to read” taxa” rather than “groups” and make clear these are the discrete units used in analysis

Caption Table 2 insert space indicates no available information

Results 115 different species or taxa collected but this is reduced to about 30 listed in Table 2. Needs more information n methods eg 6 species included in “carabid beetles” or similar
L157 add placed into X taxa for analysis
L281 earthmites (figure 3) abundance looks unaffected by treatment except for towards end of treatment? And earwigs higher abundance fr earlier sampling dates?
Insert to caption for figure 2 “..species richness insert X taxa of beneficials and pests


The relevance of species richness of all taxa collected to assessing treatment impacts need to be more clearly set out. Would like to see this analysis for “beneficials and “pests” separately. Is increased species richness desirable if it is caused by increased pest species or taxa?

Clarification needed-in Fig 3 it appears carabid abundance declines Jan 2006-June 2006 but higher rate of chlorpyrifos was applied June 21st 2006-is this correct?
L308 pest community
L309 June 2005 and September 2005 sampling but Table 3 says July 2005 and October 2005
L312 insert date of increased chlorpyrifos referred to in text to Table 3
L315staphylinids marginally affected March 2006-is this correct?
L330 and elsewhere X2 to 2 DP and P
L395 insert ref
L377 please check intent of Thomson and Hoffmann 2006 which is actually a field study not “laboratory bioassays”
L564 insert para

Figure 2 and other figures labels delete “none” and replace with “control” for consistency
Insert “X taxa of beneficials and pests” following species richness
Figure 3 caption insert units for abundance in text
Table 1 should planting dates read 2005 etc?

Table 3 insert to caption for the beneficial and pest communities

Figure 4 caption L1 and L4 shows not dhows
Label “beneficial community” not” natural enemy community” for consistency

---

## Round 0.2 · Minor Revisions

Reviewer 2 is now satisfied with revised version but reviewer 1 has some minor suggestions, which I hope you can attend to.

Reviewer 1 ·

Basic reporting

xxx

Experimental design

xxx

Validity of the findings

xxx

Additional comments

The authors have successfully improved the manuscript. Their answer to my comment on the data analysis was also convincing, but I was disappointed that they did not add at least 2-3 sentences in the manuscript like: " We are aware of xxx and therefore we conducted preliminary analyses. Theses showed that ..."

In addition a couple of minor issues should be fixed before publication:

Line numbers refer to document with tracked changes
103 add comma after chemicals
164/165 can you provide any information what these default settings are?
185 should be: the effects of pesticides
191 Pascual not in reference list - check all references carefully (and is it really Pascual or Pascoal?)
283 should be „though“ not „thought“
360 should be: „can interact“
381 should be „that reported that“ ?
406 should probably be „Stäubli“
411-413 statement of sentence is convoluted

·

Basic reporting

Clear and unambiguous, excellent English throughout with literature providing sufficient background and context

Experimental design

Research question well defined, relevant and meaningful

Validity of the findings

Data is robust statistically sound; conclusion well stated and linked to original research question

Additional comments

The authors have prepared an exemplary response to reviewer comments and the revised manuscript reflects this attention to detail

---

## Round 0.3 · accepted · Accept

Thank you for taking time to respond to reviewers' comments to improve clarity in this manuscript. I can confirm that all the comments raised by two reviewers are adequately addressed. I am happy to accept this paper for publication. Congratulations once again! I hope you will consider PeerJ for publishing your future research work.